# Distinct Clinical Phenotypes in KIF1A-Associated Neurological Disorders Result from Different Amino Acid Substitutions at the Same Residue in KIF1A

**DOI:** 10.3390/biom15050656

**Published:** 2025-05-02

**Authors:** Lu Rao, Wenxing Li, Yufeng Shen, Wendy K. Chung, Arne Gennerich

**Affiliations:** 1Department of Biochemistry and Gruss Lipper Biophotonics Center, Albert Einstein College of Medicine, Bronx, NY 10461, USA; 2Department of Systems Biology, Columbia University Irving Medical Center, New York, NY 10032, USA; 3Department of Biomedical Informatics, Columbia University Irving Medical Center, New York, NY 10032, USA; 4Department of Pediatrics, Boston Children’s Hospital, Boston, MA 02115, USA; 5Harvard Medical School, Boston, MA 02115, USA

**Keywords:** KIF1A, microtubules, genotype–phenotype associations, neurodevelopmental, neurodegenerative, KIF1A-associated neurological disorders, amino acid substitutions

## Abstract

KIF1A is a neuron-specific kinesin motor responsible for intracellular transport along axons. Pathogenic *KIF1A* mutations cause KIF1A-associated neurological disorders (KAND), a spectrum of severe neurodevelopmental and neurodegenerative conditions. While individual *KIF1A* mutations have been studied, how different substitutions at the same residue affect motor function and disease progression remains unclear. Here, we systematically examine the molecular and clinical consequences of mutations at three key motor domain residues—R216, R254, and R307—using single-molecule motility assays and genotype–phenotype associations. We find that different substitutions at the same residue produce distinct molecular phenotypes, and that homodimeric mutant motor properties correlate with developmental outcomes. In addition, we present the first analysis of heterodimeric KIF1A motors—mimicking the heterozygous context in patients—and demonstrate that while heterodimers retain substantial motility, their properties are less predictive of clinical severity than homodimers. These results highlight the finely tuned mechanochemical properties of KIF1A and suggest that dysfunctional homodimers may disproportionately drive the diverse clinical phenotypes observed in KAND. By establishing residue-specific genotype–phenotype relationships, this work provides fundamental insights into KAND pathogenesis and informs targeted therapeutic strategies.

## 1. Introduction

KIF1A is a neuron-specific member of the kinesin-3 family, a group of microtubule plus-end-directed motor proteins essential for intracellular transport [1,2]. It plays a key role in nuclear migration in differentiating brain stem cells [3,4] and facilitates the transport of synaptic precursors and dense-core vesicles to axon terminals [5,6,7,8,9,10]. Over the past decade, more than 181 pathogenic *KIF1A* variants have been identified as the cause of KIF1A-associated neurological disorders (KAND) [11,12]. KAND are highly variable and difficult to predict, encompassing a broad spectrum of neurodevelopmental and neurodegenerative conditions, including progressive spastic paraplegias, microcephaly, encephalopathies, intellectual disability, epilepsy, autism, autonomic and peripheral neuropathy, optic nerve atrophy, and cerebral and cerebellar atrophy [11,13,14,15,16,17,18,19,20,21,22,23,24,25,26,27,28,29,30,31,32,33,34,35,36,37,38,39,40,41,42,43,44,45,46,47,48,49,50,51,52,53,54].

KIF1A, like all kinesins, consists of a motor domain and a tail domain. The tail domain is highly divergent across kinesin family members, enabling functions such as oligomerization, cargo binding, and regulation [55], while the motor domain is highly conserved and serves as the catalytic core [56]. This domain binds and hydrolyzes ATP to generate force and motion for microtubule-based transport. Its ATPase activity dependents on three key elements—the P-loop, switch-1 loop, and switch-2 loop [57,58]—which coordinate ATP-Mg binding and hydrolysis in a process stimulated by microtubule interaction [59,60]. In its auto-inhibited state, full-length KIF1A is monomeric [61], with its tail domain folding back onto the motor domain to prevent dimerization [62]. Upon activation via phosphorylation [63] or cargo binding [64], KIF1A dimerizes and becomes highly processive [65].

The severity and specific manifestations of KAND depend on the location of the mutation within KIF1A and the nature of the amino acid substitution, with most pathogenic variants residing in the KIF1A motor domain [11] (Figure 1). While several *KIF1A* missense mutations have been characterized at molecular [11,66,67,68,69] or clinical level [11,12,70,71,72], the effects of different amino acid substitutions at the same residue remain poorly understood. Establishing such connections is critical for elucidating disease mechanisms and guiding potential therapeutic strategies.

To assess whether molecular phenotypes predict clinical phenotypes, we examined the molecular and clinical consequences of three distinct KAND-associated amino acid substitutions at residues R216, R254, and R307 in the KIF1A motor domain. These residues were selected because each is associated with multiple disease-linked substitutions in patients, providing a unique opportunity to assess how different amino acids variants at the same position affect motor function. Structurally, R216 and R254 are located within the switch-1 and switch-2 loops, respectively, which are key elements of the ATPase cycle, while R307 lies within a helix that directly contacts microtubules. Using a tail-truncated, dimerizing KIF1A construct (393 amino acids) expressed in *E. coli*, we investigated the effects of these mutations in single-molecule assays, as this construct recapitulates the behavior of constitutively active full-length KIF1A [68]. To approximate the physiological context of heterozygous mutations in patients, we also generated and analyzed heterodimeric KIF1A motors composed of one wild-type and one mutant motor domain.

Our findings reveal that the magnitude of the biophysical alterations induced by these mutations correlates with the severity of clinical phenotypes. Specifically, differences in residue location and amino acid properties influence KIF1A’s motility defects and, consequently, the extent of neurodevelopmental impairment. Furthermore, we show that heterodimeric motors retain partial function and exhibit more moderate motility defects, but their biophysical properties are less predictive of clinical severity than those of homodimeric mutants. These results establish a direct link between the biophysical consequences of residue-specific KIF1A mutations and their clinical impact, offering insights into disease mechanisms and potential therapeutic avenues.

## 2. Materials and Methods

### 2.1. Plasmids and Constructs

KIF1A mutants were generated from a previously described construct, KIF1A(1-393)-LZ-SNAPf-EGFP-6His [11,66,68], using Q5 mutagenesis (#E0554S, New England Biolabs, Ipswich, MA, USA). Constructs KIF1A(1-393)-RRLZ-SNAPf-EGFP-6His and KIF1A(1-393)-EELZ-HaloTag-StrepII were created for heterodimeric KIF1A. The RRLZ construct was generated by replacing the LZ with RRLZ via Gibson assembly. The EELZ construct was generated by replacing the LZ-SNAPf-EGFP-6His with EELZ and HaloTag-StrepII tag, also using Gibson assembly. All constructs were verified by Sanger sequencing (Genome Core Facility, Albert Einstein College of Medicine, Bronx, NY, USA).

### 2.2. E. coli-Based KIF1A Expression

All constructs were expressed in *E. coli*. Each plasmid was transformed into BL21-CodonPlus(DE3)-RIPL competent cells (#230280, Agilent Technologies, Santa Clara, CA, USA). A single colony was picked and inoculated in 1 mL of terrific broth (TB) [10.1101/pdb.rec8620] containing 50 µg/mL carbenicillin and 50 µg/mL chloramphenicol. The culture was shaken at 37 °C for 5 h and then cooled to 16 °C for 1 h. Protein expression was induced by 0.1 mM IPTG overnight at 16 °C. Cells were harvested by centrifugation at 3000× *g* for 10 min at 4 °C. The supernatant was discarded, and the pellet was fully resuspended in 5 mL of B-PER™ Complete Bacterial Protein Extraction Reagent (#89821, ThermoFisher Scientific, Waltham, MA, USA) supplemented with 2 mM MgCl_2_, 1 mM EGTA, 1 mM DTT, 0.1 mM ATP, and 2 mM PMSF. The resuspension was flash-frozen in liquid nitrogen and stored at –80 °C.

### 2.3. Purification of E. coli-Expressed Constructs

To purify *E. coli*-expressed protein, the frozen cell pellet was thawed at 37 °C and nutated at room temperature (RT) for 20 min to lyse the cells. For heterodimers, the two constructs were combined before nutation. The lysate was clarified by centrifugation at 80,000 rpm (260,000× *g*, *k*-factor = 28) for 10 min in an TLA-110 rotor using a Beckman Tabletop Optima TLX Ultracentrifuge. The supernatant was passed through 500 μL of Ni-NTA Roche cOmplete™ His-Tag purification resin (#5893682001, Millipore Sigma, St. Louis, MO, USA). The resin was washed with 4 mL of wash buffer (50 mM HEPES, pH 7.2, 300 mM KCl, 2 mM MgCl_2_, 1 mM EGTA, 1 mM DTT, 1 mM PMSF, 0.1 mM ATP, 0.1% (*w*/*v*) Pluronic F-127, 10% glycerol). To label the SNAPf-tag or HaloTag, SNAP-Cell^®^ TMR-Star ligand (#S9105S, New England Biolabs, Ipswich, MA, USA)—and HaloTag^®^ Alexa Fluor^®^ 660 Ligand (#G8472, Promega, Madison, WI, USA) for heterodimers—was added to the resin to a final concentration of 10 µM before elution. The resin was then incubated at RT for 20 min. The resin was then washed with 6 mL of wash buffer, and the protein was eluted using elution buffer (50 mM HEPES, pH7.2, 150 mM KCl, 2 mM MgCl_2_, 1 mM EGTA, 1 mM DTT, 1 mM PMSF, 0.1 mM ATP, 0.1% (*w*/*v*) Pluronic F-127, 10% glycerol, 150 mM imidazole). The elute was flash-frozen and stored at –80 °C. For heterodimers, the elute from the Ni-NTA resin was passed through Strep-Tactin^®^ 4Flow^®^ high-capacity resin (#2-1250-010, IBA Lifesciences GmbH, Monheim, Germany). The resin was washed with 10 mL of wash buffer (50 mM HEPES, pH 7.2, 300 mM KCl, 2 mM MgCl2, 1 mM EGTA, 1 mM DTT, 1 mM PMSF, 0.1 mM ATP, 0.1% (*w*/*v*) Pluronic F-127, 10% glycerol), and eluted with elution buffer (50 mM HEPES, pH7.2, 150 mM KCl, 2 mM MgCl2, 1 mM EGTA, 1 mM DTT, 1 mM PMSF, 0.1 mM ATP, 0.1% (*w*/*v*) Pluronic F-127, 10% glycerol, 5 mM desthiobiotin). The elute was flash-frozen and stored at –80 °C.

### 2.4. Microtubule Polymerization

To polymerize microtubules, 2 µL of 10 mg/mL tubulin (#T240-B, Cytoskeleton Inc., Denver, CO, USA) was mixed with 2 µL of 1 mg/mL biotinylated tubulin (#T333P-A, Cytoskeleton Inc.), 1 µL of 1 mg/mL HiLyte488-labeled tubulin (#TL488M-A, Cytoskeleton Inc.), and 1 µL of 10 mM GTP. The mixture was incubated at 37 °C for 20 min, followed by the addition of 0.6 µL of 0.2 mM paclitaxel in DMSO. Incubation continued for another 20 min. The polymerized microtubules were then carefully layered on top of 100 µL of a glycerol cushion (80 mM PIPES, pH 6.8, 2 mM MgCl_2_, 1 mM EGTA, 60% (*v*/*v*) glycerol, 1 mM DTT, 10 µM paclitaxel) in a 230 µL TLA100 tube (#343775, Beckman Coulter, Brea, CA, USA) and centrifuged at 80,000× *g* rpm (250,000× *g*, *k*-factor = 10) for 5 min at RT using a Beckman Tabletop Optima TLX Ultracentrifuge. The supernatant was carefully removed, and the pellet was resuspended in 11 µL of BRB80G10 (80 mM PIPES, pH 6.8, 2 mM MgCl_2_, 1 mM EGTA, 10% (*v*/*v*) glycerol, 1 mM DTT, 10 µM paclitaxel). The microtubule solution was stored at RT in the dark for further use. For the microtubule-binding and -release assay, only 5 µL of 10 mg/mL unlabeled tubulin was used for polymerization.

### 2.5. Microtubule-Binding and -Release (MTBR) Assay

A microtubule-binding and -release (MTBR) assay was performed to remove inactive motors prior to single-molecule TIRF assays. Fifty microliters of each protein construct were buffer-exchanged into a low-salt buffer (30 mM HEPES, pH 7.2, 50 mM KCl, 2 mM MgCl_2_, 1 mM EGTA, 1 mM DTT, 1 mM AMP-PNP, 10 µM paclitaxel, 0.1% (*w*/*v*) Pluronic F-127, and 10% glycerol) using a 0.5 mL Zeba™ spin desalting column (7 kDa MWCO) (#89882, ThermoFisher Scientific, Waltham, MA, USA). After buffer exchange, the solution was warmed to RT and 3 μL of 5 mg/mL paclitaxel-stabilized microtubules was added. The mixture was gently mixed and layered over a 100 μL glycerol cushion (80 mM PIPES, pH 6.8, 2 mM MgCl_2_, 1 mM EGTA, 1 mM DTT, 10 µM paclitaxel, and 60% glycerol). The sample was centrifuged at 45,000 rpm (80,000× *g*, *k*-factor = 33) for 10 min at RT using a TLA-100 rotor in a Beckman Tabletop Optima TLX Ultracentrifuge. The supernatant was carefully removed, and the pellet was resuspended in 50 μL high-salt release buffer (30 mM HEPES, pH 7.2, 300 mM KCl, 2 mM MgCl_2_, 1 mM EGTA, 1 mM DTT, 10 μM paclitaxel, 1 mM ATP, 0.1% (*w*/*v*) Pluronic F-127, and 10% glycerol). Microtubules were then removed by centrifugation at 40,000 rpm (60,000× *g*, *k*-factor = 41) for 5 min at RT. The final supernatant, containing the active motors, was aliquoted, flash-frozen in liquid nitrogen, and stored at –80 °C.

### 2.6. Total Internal Reflection Fluorescence (TIRF) Assay

#### Flow Chamber Preparation

A flow chamber was assembled using a glass slide (Fisher Scientific, #12-550-123) and an ethanol-cleaned coverslip (#474030-9000-000, Zeiss, Oberkochen, Germany) with two thin strips of parafilm as spacers. To functionalize the surface, 10 µL of 0.5 mg/mL BSA-biotin (#29130, ThermoScientific, Waltham, MA, USA) was introduced into the chamber and incubated for 10 min. The chamber was then washed with 2×20 µL blocking buffer (80 mM PIPES, pH 6.8, 2 mM MgCl2, 1 mM EGTA, 10 µM paclitaxel, 1% (*w*/*v*) Pluronic F-127, 2 mg/mL BSA, 1 mg/mL α-casein) and incubated for 30 min to block the surface. Next, 10 µL of 0.25 mg/mL streptavidin (#Z7041, Promega, Madison, WI, USA) was introduced into the chamber and incubated for 10 min, followed by another wash with 2 × 20 µL blocking buffer. Ten microliters of 0.02 mg/mL biotin-labeled microtubules in the blocking buffer were then introduced and incubated for 1 min. The chamber was washed again with 2 × 20 µL blocking buffer and stored in a humid chamber until use.

### 2.7. Sample Preparation

The motor solution was diluted to the appropriate concentration, and 1 µL of the diluted motor solution was added to 50 µL of motility buffer (80 mM PIPES, pH 7.2, 2 mM MgCl_2_, 1 mM EGTA, 10 µM paclitaxel, 0.5% (*w*/*v*) Pluronic F-127, 5 mg/mL BSA, 1 mg/mL α-casein, 2 mM ATP, 2 mM biotin, gloxy oxygen scavenger system). Two times 20 µL of the solution was introduced into the chamber, which was then sealed with vacuum grease.

### 2.8. Data Acquisition

Images were acquired using VisiView software (BioVision Technologies, Exton, PA, USA) (version 6.0.0.19) with an acquisition time of 200 ms per frame and 600 frames per movie.

### 2.9. Data Analysis

Kymographs were generated using Fiji [73], and the velocity and run length were analyzed using a home-built MATLAB program (version R2023a).

## 3. Graphic Visualization

The molecular structures in the figures were visualized using UCSF ChimeraX (version 1.9) [74] or PyMOL (version 3.1).

## 4. Clinical Studies

All studies were approved by the Institutional Review Boards at Columbia University and Boston Children’s Hospital. Informed consent was obtained from all patients or their guardians. Patient recruitment, genetic diagnosis verification, and clinical characterization were conducted as previously described [11,12]. For individuals with multiple Vineland score assessments, the most recent score was used for statistical analysis.

### 4.1. AlphaFold2 Prediction of KIF1A Variants

KIF1A mutant structures were predicted using ColabFold v1.5.5 [75].

### 4.2. AlphaMissense Scores of KIF1A Variants

AlphaMissense scores, which predict the functional impact of missense variants using a deep learning model [76], were obtained from https://github.com/google-deepmind/alphamissense (accessed on 3 January 2024) for all *KIF1A* missense variants. Higher scores indicate a greater deleterious effect on protein function.

## 5. Results

### 5.1. KIF1A Mutants Exhibit Distinct Single-Molecule Motility Behaviors

To investigate how different amino acid substitutions at the same residue affect KIF1A function, we analyzed three conserved motor domain sites: R216, R254, and R307 (Figure 1A). These residues are highly conserved or semi-conserved across kinesin families (Appendix A). R216 is located at the center of the switch-1 loop, which regulates nucleotide exchange [58]. R254 is positioned at the end of the switch-2 loop, which transmits conformational changes critical for force generation [77]. R307 lies within the conserved YPRxS motif (where x represents D, E, or N), forming a unique 3_10_-helix adjacent to loop 12 [67] (Figure 1B).

Previous studies have shown that disease-associated mutations in highly conserved motor domain residues often impair KIF1A function [11,67,68]. Here, we demonstrate that mutations at these three sites either abolish motility or significantly reduce velocity and processivity (Figure 2). Moreover, different amino acid substitutions at the same site can lead to distinct molecular behaviors, emphasizing the role of residue-specific effects in determining motor dysfunction and clinical severity.

To probe these effects, we employed total internal reflection fluorescence (TIRF) microscopy to measure single-molecule motility using a tail-truncated, dimerizing KIF1A construct (residues 1-393). Engineered with a dimerizing leucine zipper to mimic the activated state of full-length KIF1A [68], this construct exhibits a velocity of ~2.3 µm/s and a run length of ~15 µm (Figure 2A and Table 1). We systematically introduced KAND-associated mutations at R216, R254, and R307 (Figure 1A) and quantified their effects on velocity, processivity, and microtubule binding.

### 5.2. R216 Mutants Impair KIF1A Motility

R216, located within the switch-1 loop, plays a key role in KIF1A’s mechanochemical cycle by forming a salt bridge with the highly conserved E253 in the switch-2 loop (Figure 1C) [78]. The switch-1 loop undergoes substantial conformational changes during the kinesin ATPase cycle, adopting a closed conformation in the ATP-bound state and an open conformation in the ADP-bound state [66]. It has been proposed that the R216–E253 interaction functions as a “backdoor” to facilitate efficient γ-phosphate release following ATP hydrolysis [78]. However, cryo-EM structures of KIF1A bound to microtubules in AMP-PnP (a non-hydrolysable ATP analog) and ADP states suggest that this interaction remains intact in both states [66] (Figure 1C, left; Appendix A). This implies that the R216–E253 bond may break (“open the backdoor”) only when the motor domain detaches from microtubules, triggering γ-phosphate release.

Supporting this model, a recent structural study of KIF5B revealed that E236 (the equivalent of E253 in KIF1A) interacts with R203 (R216 in KIF1A) when bound to microtubules [79]. However, in the ADP state, when the motor domain is in solution, E236 instead interacts with T87 (T99 in KIF1A) of the P-loop [80], a finding corroborated by another structural study [57]. The R216–E253 interaction likely prevents premature ADP release, as mutating E236 to alanine in KIF5B leads to a 100-fold increase in ADP release rate compared to wild-type [81]. Furthermore, mutations at E236 in KIF5B or R216 in KIF1A result in significantly slower ATP hydrolysis rates [82], emphasizing the importance of this interaction in KIF1A’s ATPase cycle.

Mutations at R216 significantly impair KIF1A motility, though with distinct effects depending on the amino acid substitution (Figure 2A, middle and right). The R216C mutation completely abolishes KIF1A motility along microtubules (Figure 2A, middle), consistent with previous findings [25]. In contrast, R216H retains a low level of motility (Figure 2A, left; Table 1), likely due to the partial preservation of local interactions by the charged histidine (His). Graph-based deep learning predictions [83] suggest that while R216H does not interact with E253, it maintains interactions with the α4 helix, a region directly involved in microtubule binding of KIF1A [66].

### 5.3. R254 Mutants Retain Motility Along Microtubules

R254 is a semi-conserved residue (Appendix A) located at the end of the switch-2 loop. Its position suggests that conformational changes in the ATP-binding pocket may directly influence microtubule binding during the ATP hydrolysis cycle. Molecular modeling predicts that R254 in KIF1A (K237 in KIF5B) preferentially interacts with α-tubulin in the ATP state compared to the ADP state [84,85]. The hypothesis is supported by cryo-EM structures of KIF1A bound to microtubules in either the AMP-PnP or ADP state [66]. In the AMP-PnP state, R254 interacts with two glutamate residues (E414 and E420) on α-tubulin (Figure 1C, middle). However, in the ADP state, R254 loses this interaction and instead forms an intramolecular interaction with D339 in H6 of KIF1A (Appendix A).

Despite its strong interactions with α-tubulin in the ATP state, mutations at R254 do not abolish KIF1A’s motility (Figure 2B), suggesting that R254 facilitates, but is not essential for, motor function. Consistently with this, a previous study reported that full-length KIF1A carrying the R254Q mutations retains motility [86]. This may explain why R254 is less conserved compared to core kinesin residues, such as those within the switch loops, which have more fundamental roles in motor activity.

Interestingly, different substitutions at R254 result in distinct single-molecule phenotypes. R254Q exhibits higher velocity and longer run length than R254P and R254W (Figure 2B and Table 1), likely due to the chemical similarity between glutamine (Q) and arginine (R). While glutamine lacks arginine’s positive charge, it maintains a comparable size and hydrophilicity, making it more functionally compatible than proline or tryptophan. Structural predictions [83] suggest that R254Q partially preserves interactions with E414 on α-tubulin in the AMP-PnP state (Appendix A). In contrast, R254W disrupts the local environment due to the hydrophobicity of its aromatic ring, leading to steric clashes (Appendix A). Meanwhile, R254P, with its much smaller side chain, does not participate in significant intramolecular or intermolecular interactions, and its effect likely stems from losing microtubule interactions.

### 5.4. R307 Mutants Demonstrate Only Brief Interactions or Diffusion on Microtubules

R307 is a highly conserved residue among kinesin families, located in the 3_10_-helix following loop 12 [67] (Figure 1). Disease-associated mutations in this region, including P305L, Y306C, and R307P/Q/G, have been identified in patients with KAND [11]. Cryo-EM structures of dimeric KIF1A bound to microtubules in the presence of AMP-PnP indicate that R307 interacts with the H12 helix of β-tubulin [66]. Structural studies suggest that R278 in KIF5B (the equivalent to R307 in KIF1A) is stabilized by a salt bridge between its neighboring D279 and an arginine on the H8-S7 loop of β-tubulin, facilitating its interaction with the H12 helix [87]. This interaction is essential for kinesin’s microtubule-binding affinity.

Molecular modeling [84,85] and a structural study [66] (Figure 1C, right; Appendix A) further support R307 as a key contributor to microtubule binding in both the ATP and ADP states. A mutagenesis study on KIF5B revealed that replacing R278 (R307 in KIF1A) with alanine increases the motor’s Michaelis–Menton constant (K_m_) for microtubules more than 10-fold compared to wild-type kinesin, indicating a significant reduction in microtubule-binding affinity, while its microtubule-gliding velocity remained only slightly affected [88]. These findings suggest that mutations at this site do not abolish ATPase activity but instead severely impair microtubule attachment.

Mutations at R307 render KIF1A immobile, regardless of the substituting amino acid (Figure 2C). The mutants interact only briefly with microtubules before dissociating, with the occasional motor entering a diffusional state along the microtubule. A previous study also showed that R307P and R307Q mutants are immobile [89]. However, in that study, the mutants exhibited extensive diffusion along microtubules rather than brief interactions. This discrepancy likely arises from differences in buffer conditions: the previous study used BRB12, which has significantly lower ionic strength than BRB80, the buffer used here. Low ionic strength may enhance electrostatic interactions, promoting stronger motor–microtubule association.

The single-molecule behaviors of R307Q, R307P, and R307G are similar, with interaction durations shorter than 200 milliseconds. Unlike R254Q, where arginine-to-glutamine substitution partially preserves function, the R307 mutation does not retain motility. This underscores the importance of the salt bridge between R307 and β-tubulin in stabilizing microtubule binding.

### 5.5. Heterodimeric KIF1A Mutants Retain Motility

As pathogenic *KIF1A* variants are typically heterozygous, it is likely that ~50% of KIF1A motors in affected individuals are heterodimeric. To assess the functional impact of disease-associated mutations in this context, we characterized the function of four non-motile KIF1A homodimeric mutants (R216C, R216H, R307P, and R307Q) in a heterodimeric background. Heterodimers were generated by replacing the dimerizing coiled-coil domain of the homodimeric construct with either a highly positively charged coiled coil (RRLZ) or a highly negatively charged coiled coil (EELZ) [90]. In the EELZ construct, the C-terminal SNAPf tag was replaced with a HaloTag to enable orthogonal labeling. The WT/WT heterodimer behaved similarly to the WT homodimer, exhibiting only a slight reduction in processivity (Figure 3 and Table 1), confirming that the heterodimeric system is a reliable tool for probing the effects of KIF1A motor domain mutations. In contrast to the immotile behavior of the corresponding homodimers (Figure 2), heterodimers containing a WT motor and a mutant heavy chain (R216C, R216H, R307P, or R307Q) were motile, though with reduced performance. These heterodimers showed ~40% lower velocity and ~20–30% reduced processivity compared to WT (Figure 3 and Table 1). These results indicate that heterodimeric KIF1A motors retain substantial motility when paired with a WT motor domain, despite the presence of a non-functional partner domain.

### 5.6. Mutant KIF1A Motor Properties Correlate with Clinical Outcomes

To examine the relationship between KIF1A motor function and clinical severity, we analyzed the Vineland Adaptive Behavior Scale (VABS) Adaptive Behavior Composite (ABC), a composite quantitative score of motor and adaptive function normalized for age. The VABS ABC score integrates performance across communication, daily living skills, and socialization domains. It is valid from birth to 90 years and widely used in the clinical evaluation of developmental disorders, with a population mean of 100 and a standard deviation of 15.

We focused on individuals carrying heterozygous mutations at three recurrent residues in KIF1A—R216, R254, and R307—which were represented in our cohort by 9, 28, and 9 individuals, respectively. Motor properties of the corresponding homodimeric KIF1A mutants (velocity, processivity, and diffusion)—expected to reflect ~25% of the molecular motor population in heterozygous individuals—were significantly correlated with VABS ABC scores (Table 2). Individuals with R307 mutations exhibited markedly lower VABS ABC scores than those with R254 (mean: 39 vs. 65; *p* = 0.001) or R216 mutations (mean: 39 vs. 64; *p* = 0.003), consistent with the near-complete immotility of the R307 homodimeric mutants observed in our single-molecule assays (Figure 2). In contrast, R216 and R254 mutants retained partial motility (Figure 2).

Within the R254 group, individuals with the R254Q variant had marginally higher VABS ABC scores than those with R254P (76 vs. 58; *p* = 0.04) or R254W (76 vs. 63; *p* = 0.11) (Table 3). This trend parallels our in vitro data showing that the homodimeric R254Q mutant exhibits slightly higher velocity and processivity compared to the other R254 variants, R216, and R307 mutants.

To determine whether heterodimeric KIF1A motors also show genotype–phenotype relationships, we analyzed the clinical data for individuals carrying R216C, R216H, R307P, or R307Q mutations—variants for which corresponding heterodimeric motility data were available. In contrast to the homodimeric analysis, no significant association was observed between VABS ABC scores and the motility properties of the heterodimeric mutants (Table 2). Nonetheless, the heterodimeric R307 mutants remained less processive than their R216 counterparts, mirroring the molecular trends seen in the homodimeric background.

Together, these findings support a strong relationship between motor dysfunction and clinical phenotype, particularly for homodimeric KIF1A mutants. The observed correlation between reduced motility and lower VABS ABC scores suggests that dysfunctional homodimers—expected to constitute ~25% of KIF1A motors in heterozygous individuals—may disproportionately drive disease severity. These results indicate that reduced motor activity in this subpopulation of KIF1A dimers is sufficient to contribute measurably to neurodevelopmental impairment.

## 6. Discussion

KIF1A plays an essential role in intracellular transport by delivering synaptic vesicle precursors (SVPs) and dense-core vesicles along axons in neurons [5,6,7,8,9,10]. Given its critical function in neuronal health, pathogenic mutations in *KIF1A* lead to a spectrum of neurodevelopmental and neurodegenerative conditions collectively known as KIF1A-associated neurological disorders (KAND) [11]. Understanding how specific mutations disrupt KIF1A function is key to uncovering disease mechanisms and developing targeted therapies. While prior studies have characterized the impact of individual KIF1A mutations at the single-molecule level, the effect of different amino acid substitutions at the same residue has remained unexplored.

Here, we systematically investigated three KAND-associated mutation sites—R216, R254, and R307—within the KIF1A motor domain [11] to determine how distinct substitutions at each site alter motor function. Although AlphaFold2 [75] structural predictions suggest that these mutations do not disrupt the overall fold of the motor domain (Appendix A), and AlphaMissense predicts all substitutions to be highly deleterious [76], our single-molecule assays reveal striking functional diversity. These findings underscore the value of biophysical measurements in revealing context-specific consequences that are not captured by structural models or conservation-based predictions.

Mutations at R216, located within the switch-1 loop, produced severe motility defects that varied by substitution. R216C abolished motility, while R216H retained limited processive movement, likely due to the preservation of local contacts. This residue forms a salt bridge with E253 in switch-2 [78], a regulatory interaction implicated in efficient ATP hydrolysis and microtubule binding [78]. Our findings reinforce the mechanistic importance of this interaction and suggest that functional compensation may occur when partial structural integrity is maintained.

In contrast, substitutions at R254, a semi-conserved residue at the end of switch-2, affected motor performance in a more graded fashion. R254Q—chemically most similar to the wild-type arginine—preserved the highest motility, while R254P or R254W showed more severe defects. Structural modeling suggests that R254Q maintains partial interactions with α-tubulin, whereas R254W introduces steric hindrance and R254P results in a loss of stabilizing interactions. These results suggest that R254 facilitates, but is not essential for, microtubule-based motility, making this site more tolerant to substitution.

All mutations at R307, a highly conserved residue in the 3_10_-helix following loop 12 [67], resulted in uniform and severe functional impairment. R307 mutants exhibited only brief microtubule interactions or diffusion, with no sustained motility. Structural studies implicate R307 in stabilizing KIF1A’s interaction with the β-tubulin H12 helix in both the ATP and ADP states [66,84,85]. The disruption of this contact likely underlies the loss of microtubule-binding affinity and complete functional collapse observed in all R307 variants, consistent with previous findings for the equivalent residue in KIF5B [88].

Importantly, we found that the motility phenotypes of these homodimeric mutants strongly correlate with clinical severity in KAND patients. Individuals carrying R307 mutations had significantly lower VABS ABC scores than those with R216 or R254 mutations, which is consistent with the complete immotility of the R307 mutants. Within the R254 group, individuals with the R254Q substitution had milder impairments than those with R254W or R254P, reflecting the graded differences in motor activity. These findings provide a direct mechanistic link between specific molecular motor defects and neurodevelopmental outcomes and suggest that even modest reductions in KIF1A motility can have substantial clinical consequences.

To better approximate the physiological context of heterozygous mutations in KAND patients, we also analyzed the motility of four representative KIF1A mutants (R216C, R216H, R307P, and R307Q) in heterodimeric motors composed of one mutant and one wild-type motor domain. Unlike their homodimeric counterparts, all heterodimeric mutants retained motility, though with reduced velocity and processivity relative to WT/WT controls. These findings suggest that co-assembly with a WT motor domain partially compensates for the functional deficits of the mutant head. Notably, the heterodimeric R307 mutants remained less processive than their R216 counterparts, mirroring the more severe defects seen in the homodimeric assays.

Interestingly, the motility differences observed among heterodimeric constructs did not correlate significantly with clinical severity. This may indicate that the functional heterodimers (~50%) contribute to a shared baseline of KIF1A transport activity in patients, while the non-functional homodimers (~25%)—which vary more dramatically in severity—are drivers of the clinical heterogeneity. In other words, while heterodimers help preserve a minimal level of function across genotypes, the presence and severity of functionally impaired homodimers may measurably determine the differences in clinical outcomes observed in KAND.

Together, these results highlight the importance of residue-specific and context-dependent effects in shaping KIF1A motor activity. The distinct functional consequences of different substitutions at R216 and R254 contrast with the uniform loss-of-function seen at R307, underscoring the highly optimized nature of KIF1A’s mechanochemical cycle. Moreover, our data suggest that both homodimeric and heterodimeric motors contribute to disease pathogenesis in KAND but homodimer dysfunction may be the key determinant of clinical severity. These insights offer a compelling framework for developing mutation-specific therapeutic strategies and support the use of single-molecule motility assays in guiding precision medicine for KAND.

## 7. Conclusions

This study demonstrates that different amino acid substitutions at the same KIF1A residue are associated with distinct biophysical and clinical outcomes. By analyzing the single-molecule motility of KAND-associated mutations at R216, R254, and R307, we show that both the structural context of the residue and the specific properties of the substituted amino acid determine the extent of motor dysfunction.

Importantly, these mutation-specific differences in KIF1A motility correlate with neurodevelopmental severity in patients, establishing a direct link between molecular mechanism and clinical phenotype. Our results also reveal that while heterodimeric motors retain partial function in the presence of a wild-type head, dysfunctional homodimers may be the primary contributors to the broad spectrum of clinical presentations observed in KAND.

These findings highlight the finely tuned nature of KIF1A’s mechanochemistry and underscore the value of residue-level resolution in understanding genotype–phenotype relationships. This work lays a foundation for future precision medicine approaches aimed at tailoring therapies to the specific functional consequences of individual mutations.

## Figures and Tables

**Figure 1 biomolecules-15-00656-f001:**
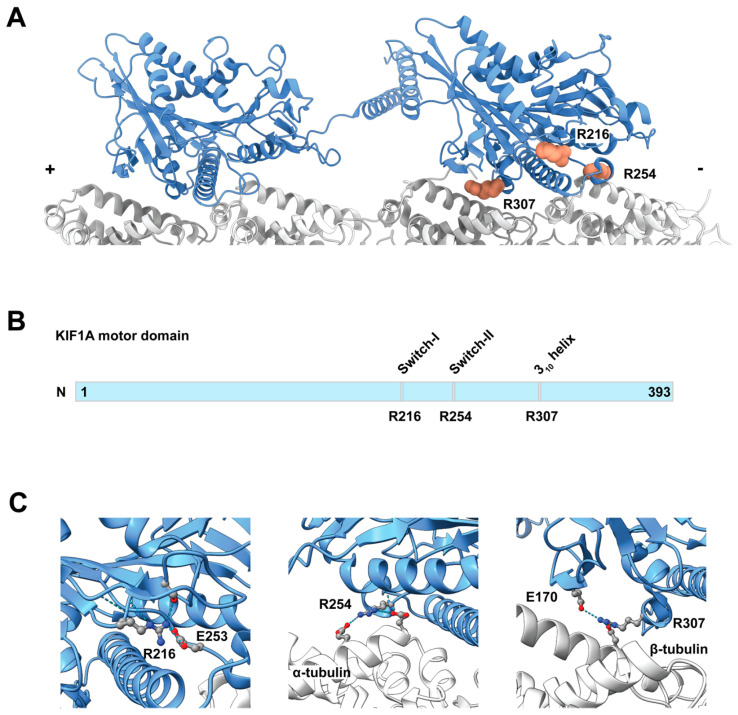
(**A**) Structure of a dimeric KIF1A bound on microtubules in the AMP-PnP state (PDB 8UTN) [66]. Residues R216, R254, and R307 in the trailing head are shown in pink in a space-filling representation. (**B**) Scheme of the KIF1A motor domain (amino acids 1-393) highlighting the locations of the three residues. (**C**) Close-up views of residue interactions: (Left) R216 interacts with S214 in the switch-I loop and E253 in the switch-2 loop. (Middle) R254 interacts with E414 and E420 in α-tubulin. (Right) R307 interacts intramolecularly with E170 and with D417 in β-tubulin.

**Figure 2 biomolecules-15-00656-f002:**
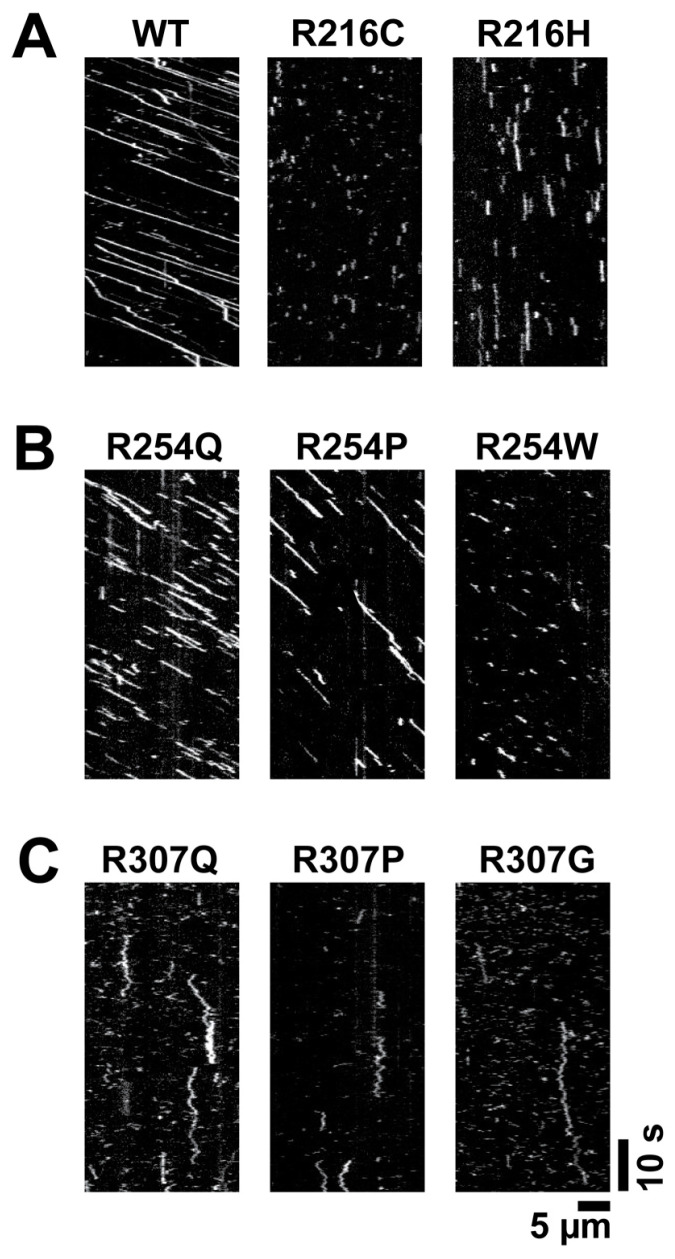
Representative kymographs of KIF1A mutants. (**A**) Wild-type (WT) KIF1A (left), R216C (middle), and R216H (right). (**B**) R254Q, R254P, and R254W. (**C**) R307Q, R307P, and R307G.

**Figure 3 biomolecules-15-00656-f003:**
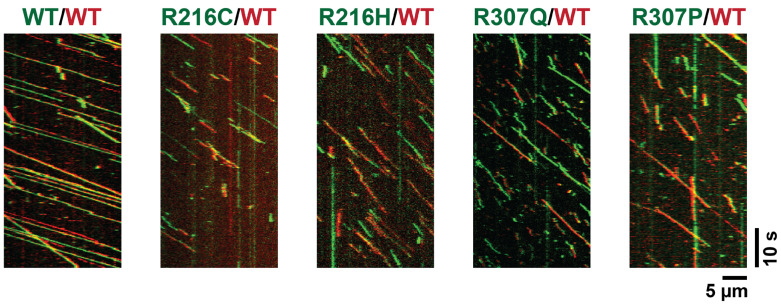
Kymographs of heterodimeric KIF1A mutants paired with WT motors. For the WT/WT heterodimer, the KIF1A motor domain fused to a SNAPf-tag was labeled with SNAP-TMR (green), and the motor domain fused to a HaloTag was labeled with Halo-Alexa660 (red). In heterodimers containing a mutant KIF1A head, the mutant was labeled with SNAP-TMR (green), and the WT head was labeled with Halo-Alexa660 (red).

**Table 1 biomolecules-15-00656-t001:** Motility of KIF1A WT and mutants. The values represent median value with 95% confidence interval. WT: n = 388; R216C: n = 214; R216H: n = 263; R254Q: n = 304; R254P: n = 354; R254W: n = 235; WT/WT: n = 422; R216C/WT: n = 477; R216H/WT: n = 494; R307Q/WT: n = 377; R307P/WT: n = 419.

	Velocity (µm/s)	Processivity (µm)	Dwell Time (s)
**Homodimers**			
WT	2.34 [2.23, 2.36]	14.5 [13.3, 15.6]	6.4 [5.9, 6.8]
R216C	-	-	0.97 [0.89, 1.04]
R216H	0.096 [0.091, 0.107]	0.46 [0.44, 0.49]	4.7 [4.5, 5.2]
R254Q	1.43 [1.40, 1.47]	2.3 [2.0, 2.5]	1.6 [1.5, 1.8]
R254P	0.80 [0.78, 0.81]	1.7 [1.4, 1.8]	2.1 [1.9, 2.3]
R254W	1.09 [1.05, 1.14]	1.2 [1.1, 1.4]	1.2 [1.0, 1.2]
R307Q	-	-	<0.2
R307P	-	-	<0.2
R307G	-	-	<0.2
**Heterodimers**			
WT/WT	2.29 [2.24, 2.34]	11.1 [10.1, 12.6]	5.0 [4.6, 5.6]
R216C/WT	1.13 [1.12, 1.15]	3.1 [2.8, 3.6]	2.7 [2.5, 3.1]
R216H/WT	0.82 [0.80, 0.83]	3.0 [2.7, 3.4]	4.0 [3.4, 4.3]
R307Q/WT	0.80 [0.75, 0.80]	1.4 [1.4, 1.6]	1.9 [1.8, 2.0]
R307P/WT	0.87 [0.83, 0.90]	2.7 [2.4, 3.0]	3.3 [2.9, 3.8]

**Table 2 biomolecules-15-00656-t002:** Univariate linear regression of key molecular phenotypes and VABS ABC.

Variable	N	Mean ± SD	Beta (SE)	P
**Homodimers**				
Diffusion				
No	28	65.14 ± 11.07	Reference	
Yes	18	51.22 ± 19.12	−13.92 (4.44)	0.003
Velocity	46		14.18 (3.80)	0.001
Processivity	46		11.80 (2.82)	<0.001
Dwell time	46		3.40 (1.74)	0.058
**R216** **C/H**				
Diffusion				
No	0	-	Reference	
Yes	9	63.56 ± 12.40	-	-
Velocity	9		−31.67 (92.97)	0.743
Processivity	9		−6.88 (20.21)	0.743
Dwell time	9		−0.85 (2.49)	0.743
**R254 P/Q/W**				
Diffusion				
No	28	65.14 ± 11.07	Reference	
Yes	0	-	-	-
Velocity	28		33.19 (11.79)	0.009
Processivity	28		10.87 (4.55)	0.025
Dwell time	28		5.07 (8.11)	0.537
**R307 G/P/Q**				
Diffusion				
No	0	-	Reference	
Yes	9	38.89 ± 16.75	-	-
Velocity	9		-	-
Processivity	9		-	-
Dwell time	9		-	-
**Heterodimers**				
Velocity	17	46.80 (39.08)		0.250
Processivity	17	10.67 (5.88)		0.090
Dwell time	17	6.83 (5.21)		0.210
**R216** **C/H**				
Velocity	9		10.22 (29.99)	0.743
Processivity	9		31.67 (92.97)	0.743
Dwell time	9		−2.44 (7.15)	0.743
**R307 P/Q**				
Velocity	8		−321.43 (164.51)	0.099
Processivity	8		−17.31 (8.86)	0.099
Dwell time	8		−16.07 (8.23)	0.099

**Table 3 biomolecules-15-00656-t003:** Summary of VABS ABC in patients with heterozygous *KIF1A* mutations.

Protein Notation	N	Mean ± SD
R216	9	63.56 ± 12.40
R216C	3	65.67 ± 5.13
R216H	6	62.50 ± 15.22
R254	28	65.14 ± 11.07
R254P	2	58.00 ± 2.83
R254Q	5	76.40 ± 14.60
R254W	21	63.14 ± 9.02
R307	9	36.88 ± 16.70
R307G	1	55.00
R307P	2	20.00
R307Q	6	42.50 ± 15.45

## Data Availability

Data supporting the findings of this manuscript are available from the corresponding author upon request.

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
