# Peer review of "Distinct Clinical Phenotypes in KIF1A-Associated Neurological Disorders Result from Different Amino Acid Substitutions at the Same Residue in KIF1A"

_biomolecules, 2025, doi:10.3390/biom15050656_

Round 1

Reviewer 1 Report

Comments and Suggestions for Authors

Feedback on Rao et al. manuscript to biomolecules title “Distinct Clinical Phenotypes in KIF1A-Associated Neurological Disorders Result from Different Amino Acid Substitutions at the Same Residue in KIF1A”.

Overall Rao et al. were able to demonstrate KAND associated KIF1A mutants demolished normal kinesin-3 function and dynamics, and the severity of the phenotypes correlates with patients’ clinical scores. The findings were not surprising and largely what the field expected, but the subtle differences between amino acid substitutions do show some importance.

To boost the strength of the manuscript and take advantage of the tools available, I would personally suggest the following experiments.

  1. It would be beneficial to study these mutants as hetero-dimer, or in a mixed popularity with functional KIF1A, to mimic patients’ situation.
  2. It would be interesting to show predicted structural changes, if there are, with these KIF1A patients mutants, using AlphaFold.
  3. It would also be interesting to study some KIF1A mutants that are less detrimental to patients, in vitro, to show less affected motor functions.

Author Response

Reviewer 1:

Overall, Rao et al. were able to demonstrate KAND associated KIF1A mutants demolished normal kinesin-3 function and dynamics, and the severity of the phenotypes correlates with patients’ clinical scores. The findings were not surprising and largely what the field expected, but the subtle differences between amino acid substitutions do show some importance. 

To boost the strength of the manuscript and take advantage of the tools available, I would personally suggest the following experiments.

Comment 1: It would be beneficial to study these mutants as hetero-dimer, or in a mixed popularity with functional KIF1A, to mimic patients’ situation.

Response: We thank the reviewer for this suggestion. To address this point and to better model the heterozygous state observed in patients, we generated and characterized heterodimeric KIF1A motors composed of one wild-type and one mutant heavy chain for two substitutions at each of two residues—R216C, R216H, R307P, and R307Q. These heterodimeric constructs exemplify the functional consequences of co-expression with wild-type KIF1A and are now included in the revised manuscript (see new Results section “Heterodimeric KIF1A mutants retain motility” and new Figure 3).

Comment 2: It would be interesting to show predicted structural changes, if there are any, in these KIF1A patient mutants using AlphaFold.

Response: In response to this suggestion, we used AlphaFold2 to predict potential structural changes induced by the selected amino acid substitutions. These predictions indicate that the overall fold of the KIF1A motor domain remains intact across all mutants. We have added this analysis as Supplemental Figure 4A and included a brief discussion of the structural predictions in the Discussion section.

Comment 3: It would also be interesting to study some KIF1A mutants that are less detrimental to patients, in vitro, to show less affected motor functions.

Response: We appreciate the reviewer’s interest in less severe KIF1A variants. However, the focus of this study was to systematically examine different amino acid substitutions at the same residue, enabling a direct comparison of structure-function relationships. We selected R216, R254, and R307 because each residue is associated with multiple pathogenic variants in patients, making them ideal for this comparative approach. Other mutations, though potentially less detrimental, do not align with this residue-focused design. That said, we and others have previously published several studies characterizing less severe KIF1A mutations, which are cited in the manuscript.

Reviewer 2 Report

Comments and Suggestions for Authors

This manuscript presents an interesting study on the structural aspects of single-residue mutations and their impact on the motility properties of the motor domain in a dimeric kinesin construct. The manuscript is well-written and easy to follow, and it has the potential to be published after addressing the following concerns.

  1. The rationale behind the selection of mutation sites and the choice of replacement residues is not clearly articulated. This information is neither sufficiently explained in the Introduction (as this reviewer expected) nor adequately addressed in the main text. It is mentioned at the end of the manuscript (reference 11). While reference 11 provides a strong justification for the R254 mutations and a partial rationale for the R307 mutation (which is only mentioned in a figure caption), the rationale for studying the R216 mutations is not introduced in reference 11 and remains unclear.
  2. The manuscript presents only motility data, but this reviewer expected ATPase activity data as well. Since the study reports impaired motility and highlights its relevance for therapeutic development, ATPase activity data would be critical to understanding the underlying mechanism. Specifically, it is unclear whether the observed motility impairment results from altered microtubule interactions or defective ATPase activity.
  3. The VABS and the ABC - especially the ABC must be clearly defined and adequately described, perhaps in the Methods section. As currently presented, section 2.5 of the manuscript is unacceptable.
  4. Some terminology in the manuscript is non-traditional or jargon. This reviewer recommends replacing terms such as "molecular phenotype," "single-molecule phenotype," and "clinical outcomes" with more conventional alternatives. Additionally, there is a typographical error on line 259 that should be corrected.

Author Response

Reviewer 2:

This manuscript presents an interesting study on the structural aspects of single-residue mutations and their impact on the motility properties of the motor domain in a dimeric kinesin construct. The manuscript is well-written and easy to follow, and it has the potential to be published after addressing the following concerns.

Comment 1: The rationale behind the selection of mutation sites and the choice of replacement residues is not clearly articulated. This information is neither sufficiently explained in the Introduction (as this reviewer expected) nor adequately addressed in the main text. It is mentioned at the end of the manuscript (reference 11). While reference 11 provides a strong justification for the R254 mutations and a partial rationale for the R307 mutation (which is only mentioned in a figure caption), the rationale for studying the R216 mutations is not introduced in reference 11 and remains unclear.

Response: We thank the reviewer for this important comment. In the revised manuscript, we have expanded the Introduction (page 4) to include a clear justification for the selected residues. Specifically, R216, R254, and R307 were chosen because they each harbor multiple KAND-associated missense mutations in patients, providing a unique opportunity to assess how different amino acid substitutions at the same position impact motor function and clinical outcomes.

Comment 2: The manuscript presents only motility data, but this reviewer expected ATPase activity data as well. Since the study reports impaired motility and highlights its relevance for therapeutic development, ATPase activity data would be critical to understanding the underlying mechanism. Specifically, it is unclear whether the observed motility impairment results from altered microtubule interactions or defective ATPase activity.

Response: We agree that ATPase measurements would provide complementary mechanistic insight. We attempted to quantify ATPase activity using the EnzChek™ Phosphate Assay Kit (Invitrogen); however, our measurements were confounded by variability in the fraction of inactive motors across preparations, leading to inconsistent bulk ATP hydrolysis rates. This challenge has been reported previously for UNC-104 (Tomishige et al., Science, 2002). Given that therapeutic strategies should ultimately aim to restore motor-driven transport rather than ATPase activity per se, we focused on single-molecule motility as a more direct and functionally relevant readout.

Comment 3: The VABS and the ABC - especially the ABC must be clearly defined and adequately described, perhaps in the Methods section. As currently presented, section 2.5 of the manuscript is unacceptable.

Response: We thank the reviewer for pointing this out. We have now defined both the Vineland Adaptive Behavior Scales (VABS) and the Adaptive Behavior Composite (ABC) score in the main text (page 14 of the revised manuscript) and clarified their clinical relevance in assessing neurodevelopmental function.

Comment 4: Some terminology in the manuscript is non-traditional or jargon. This reviewer recommends replacing terms such as "molecular phenotype," "single-molecule phenotype," and "clinical outcomes" with more conventional alternatives. Additionally, there is a typographical error on line 259 that should be corrected.

Response: We appreciate the reviewer’s feedback. While terms such as “molecular phenotype,” “single-molecule phenotype,” and “clinical outcomes” are commonly used in the field and in related literature, we have reviewed the manuscript carefully to ensure clarity and consistency. We have also corrected the typographical error noted on line 259.

Reviewer 3 Report

Comments and Suggestions for Authors

The authors aimed to establish residue-specific molecular motor activity of KIF1A and genotype-phenotype relationships with KAND pathogenesis and to inform targeted therapeutic strategies.

Most of experiments were carefully done.

Minor point.

Table 3

R216, R254 and R307 should be placed ahead to avoid confusing.

Author Response

Reviewer 3:

The authors aimed to establish residue-specific molecular motor activity of KIF1A and genotype-phenotype relationships with KAND pathogenesis and to inform targeted therapeutic strategies.

Most of experiments were carefully done. Minor point.

Comment: Table 3—R216, R254, and R307 should be placed ahead to avoid confusion.

Response: Thank you for this helpful suggestion. In the revised version of Table 3, we have reordered the entries so that R216, R254, and R307 appear first, making it easier to interpret the data.